# Platelet rich plasma versus placebo for the management of Achilles tendinopathy: protocol for the UK study of Achilles tendinopathy management (ATM) multi-centre randomised trial

Rebecca Samantha Kearney ,[1] Nicholas Parsons,[2] Chen Ji,[1] Jane Warwick,[1] Jaclyn Brown,[1] Jonathan Young,[3] Matthew L Costa[4]

For numbered affiliations see end of article.

**Correspondence to**
Dr Rebecca Samantha Kearney; R.S.Kearney@Warwick.ac.uk

## ABSTRACT

**Introduction** In the UK, 150 000 people every year experience mid-substance Achilles tendinopathy. Typically patients are offered a range of treatment options such as exercise, electrotherapy, injections and surgery. With large variations in current practice, there is a pressing need to establish which treatments are effective and which are not. This is the protocol for a multi-centre randomised trial of platelet rich plasma (PRP) versus placebo injection for patients with Achilles tendinopathy.

**Methods and analysis** Adult patients with mid-substance Achilles tendinopathy for longer than 3 months will be screened. Randomisation will be on a 1:1 basis, stratified by centre and bilateral presentation. Participants will be allocated to either a single PRP injection or placebo injection. A minimum of 240 patients will be recruited into the study; this number will provide 90% power to detect a difference of 12 points in Victorian Institute of Sport Assessment-Achilles score at 6 months. Quality of life, pain and complications data will be collected at baseline, 2-week, 3-month and 6-month post-randomisation. The differences between treatment groups will be assessed on an intention-to-treat basis.

**Ethics, registration and dissemination** This trial was funded by Versus Arthritis and commenced on 1 September 2015 (Versus Arthritis 20831). National Research Ethic Committee approved this study on 30 October 2015 (15/WM/0359). It was registered on the International Standard Randomised Controlled Trial Number (ISRCTN) registry with reference number ISRCTN 13254422 on 28 October 2015. The first site opened to recruitment on 27 April 2016 and the trial was in active recruitment at the point of submitting the protocol paper. The results of this trial will be submitted to a peer-reviewed journal and will inform clinical practice with regard to the treatment of Achilles tendinopathy.

## INTRODUCTION

Tendinopathy in the mid-substance of the Achilles tendon occurs because of the failure of the tendon to mediate its repair and degeneration processes.[1] The general population

### Strengths and limitations of this study

► Blinding of interventions.
► UK wide trial across a minimum of 20 centres to optimise external validity.
► Primary outcome measure is patient-centred.
► The main limitation is that cost effectiveness analysis was not included.

has an incidence of 2.35 per 1000 people, equivalent to approximately 150 000 people in the UK every year.[2] Achilles tendinopathy is characterised by pain and stiffness over the lower portion of the calf (mid-portion of the Achilles) impacting on all weight bearing activities.[3]

Patients are offered a range of non-operative treatment options for mid-substance Achilles tendinopathy including exercise, electrotherapy and injections. Of these platelet rich plasma (PRP) injections have gained national and international interest following guidelines published by the UK National Institute for Health and Care Excellence (NICE) in 2009 (updated 2013)[4] and international guidance published by the International Olympic Committee 2010.[5] Both have highlighted PRP injections as a priority area for research, on the basis that they have the potential to reduce morbidity and the need for surgery. With large variations in current practice, there is a pressing need to establish which non-operative treatments are effective.[6–8]

PRP is defined as the plasma fraction of a patient's own blood (autologous blood) having a platelet concentration above baseline. Platelets contribute to healing through the action of growth factors.[9] A Cochrane review of injection therapies for Achilles

tendinopathy was published in 2015[10] and identified only two randomised controlled trials (RCTs) evaluating PRP. The first of these was a feasibility RCT for this study that recruited 20 participants at a single site,[3] and the second was an RCT that investigated the incremental benefit of adding PRP injections to eccentric loading exercises for 54 participants at a single site.[11] Following this review two further RCTs were published in 2016 and 2017.[12 13] They evaluated 60 and 24 participants and both were single site and compared PRP and eccentric loading to placebo and eccentric loading. Of these four trials only one[12] has documented a significant treatment effect of PRP at 6 months (mean Victorian institute of Sport Assessment-Achilles (VISA-A) score 19.6 (SD=4.5) in PRP group and 8.8 (SD=3.3) in placebo group).

Currently available RCT evidence for mid-substance Achilles tendinopathy is limited to studies recruiting at a single site with small sample size (<60 in each study and only 158 participants in total across all studies) that only evaluate the incremental effect of PRP in addition to pre-specified eccentric exercise based programmes. No studies have evaluated the effectiveness of PRP across multiple sites, in an adequately powered trial. There is a pressing need for adequately powered, multi-centre trials in this important clinical area, which this project will address.

## PREPARATORY FEASIBILITY RESEARCH

Between May 2009 and March 2012, our research group undertook a pilot RCT, funded by the Chartered Society of Physiotherapy, to evaluate the feasibility of a main RCT.[3] This study used a process evaluation model to determine the feasibility and acceptability of trial procedures. This work was completed in consultation with a patient user panel and was presented at a Versus Arthritis Achilles tendon think tank in April 2013 to further refine the trial question and procedures.

## RESEARCH QUESTION

In adults with painful mid-substance Achilles tendinopathy lasting longer than 3 months, does a single injection of PRP (intra-tendinous injection with a peppering technique) improve VISA-A scores by a minimum of 12 points when compared with a placebo (subcutaneous injection of a dry needle and no peppering technique) injection at 6-month post-injection?

## OBJECTIVES
### Primary objective
To quantify and draw inferences on observed differences in the VISA-A score between the trial treatment groups at 6-month post-randomisation.

### Secondary objectives
1. To quantify and draw inferences on observed differences in the VISA-A score at 3-month post-randomisation.

2. To identify any differences in health-related quality of life measurement (HRQoL) between trial treatment groups at 3-month and 6-month post-randomisation.
3. To identify any differences in pain measurements between trial treatment groups at 2-week, 3-month and 6-month post-randomisation.
4. To report the complication profile of PRP injections at 2-week, 3-month and 6-month post-randomisation.

## METHODS AND ANALYSIS
### Study design
A single-participant blinded, multi-centre, randomised placebo controlled trial. (online supplementary file 1).

### Sample size
The primary outcome for the study is the VISA-A score[14]; it has a range between 0 and 100, a lower score indicating more symptoms and greater limitation of physical activity and 100 being asymptomatic. The minimum clinically important difference for the VISA-A score is estimated to lie between 10 and 12 points.[11 12] From our previous feasibility publication,[3] VISA-A scores were observed to be approximately normally distributed with a SD of 26. If the true difference between the experimental and control treatment group means is 12 points, a sample of 100 patients in each group will be required to reject the null hypothesis (population means of the experimental and control groups are equal) with probability 0.9 (90% power). This equates to an effect size of 0.46, which we would consider to be moderate. The type I error rate (significance level) associated with this test is 5%. Allowing approximately 15% loss to follow-up, this amounts to 240 patients in total.

### Outcome measures
Outcome measures will be collected at baseline (pre-randomisation) by a suitably qualified member of the research team, face to face at the recruiting site. Baseline participant characteristics to be collected include duration of symptoms, side affected (left, right, bilateral), previous treatments for the Achilles tendinopathy, current medications, weight, height, age, gender, ethnicity, smoking and drinking status and employment status. Participants will then be followed up at 2 weeks by telephone direct from Warwick Clinical Trials Unit (WCTU) and 3-month and 6-month post-randomisation by postal questionnaire sent from WCTU or telephone if no postal response is received (online supplementary file 2).

### Primary
Currently the VISA-A questionnaire is the only patient reported outcome measure developed with supporting validation and reliability evidence, for this common musculoskeletal presentation.[14] The VISA-A is condition specific and is designed to have greater sensitivity and specificity than general purpose scales in the target population. The VISA-A contains eight questions that cover three domains of pain, function and activity. An

asymptomatic person would score 100, the lower the score the greater the disability. It has been shown to have good test-re-test reliability (r=0.93), inter-rater and intra-rater reliability (r=0.90) and construct validity when the mean scores were compared across patient populations with differing ranges of severity, with no evidence of floor or ceiling effects.[15] These data will be collected at baseline 3 and 6 months after randomisation, where the 6-month score is the primary outcome.

## Secondary

Euro-Qol 5 Dimensions Score (EQ-5D-5L[16]) is a generic, validated, cross-disciplinary standardised health utility instrument widely used to assess HRQoL. It has two parts, a visual analogue scale (VAS), which measures self-rated health, and a health status instrument, which will be the focus of this study, consisting of a five-level response (no problems, slight problems, moderate problems, severe problems and extreme problems) for five health domains related to daily activities: (i) mobility, (ii) self-care, (iii) usual activities, (iv) pain and discomfort and (v) anxiety and depression. The five dimensions are combined together to provide a 5-digit number that describes the individual's health state. The 5-digit responses, from the EQ-5D health classifications, are converted into an overall score using a published utility algorithm for the UK population.[17] It has good test-retest reliability, is simple for participants to use and gives a single preference-based index value for health status that can be used for broader cost-effectiveness comparative purposes.[18] These data will be collected at baseline, 3 and 6 months after randomisation.

Pain is assessed using a VAS comprising of a horizontal 10 cm line in length, anchored by two verbal descriptors (no pain and worst imaginable pain) to indicate a participant's pain level on the day on completing the questionnaire.[19] These data will be collected at baseline 2 weeks, 3 and 6 months after randomisation.

Participants will be asked if they have experienced any complications during follow-up data collection at 2 weeks, 3 and 6 months after randomisation. Complications expected and related to the study treatment that are collected at the specified time points consist of bruising and discomfort at the venesection site, syncopal (fainting) episode associated with venesection or tendon injection, infection, mild discomfort and bleeding at the injection site, swelling, skin discolouration and allergic reaction.

## Screening and eligibility

Eligible patients will be identified from foot and ankle clinics by the local principal investigator (PI) and invited to speak to a suitably qualified member of the research team. The research team member will complete the eligibility checklist in conjunction with a suitably qualified member of the clinical team. Screening logs will be collected by the clinical care team in a minimum of 20 National Health Service trusts in the UK to assess the main reasons for patient exclusion as well as the number of patients willing to take part.

Patients will be provided with verbal and written information about the study. Written informed consent will be obtained (online supplementary file 3) by a suitably qualified member of the research team, after allowing sufficient time for the patient to consider their decision and ask questions about the trial.

Patients aged 18 years or over with pain at the mid-substance of the Achilles tendon for longer than 3 months confirmed by ultrasound and/or MRI are eligible for the trial. Patients will be excluded for the following reasons: (i) if there is presence of a systemic conditions (including diabetes, rheumatoid arthritis, peripheral vascular disease), (ii) if they are unable to adhere to trial procedures, are pregnant or actively trying to become pregnant or breastfeeding at the time of randomisation, (iii) if they have had prior Achilles tendon surgery or rupture on the index side, have had previous major tendon or ankle injury or deformity to either lower leg, (iv) if they have had a fracture of a long bone in either lower limb in the previous 6 months, (v) if they have previously been randomised in the study, (vi) if they have previously had PRP treatment into a tendon or have any contraindication to receiving a PRP injection (haemodynamic instability, platelet dysfunction syndrome, active cancer, septicaemia, systemic use of anticoagulant therapy (eg, warfarin, dabigatran, heparin), local infection at site of the procedure). Low dose aspirin use (or equivalent) is not a contraindication to receiving a PRP injection; therefore, these patients will not be excluded from the study.

Patients presenting with bilateral Achilles tendinopathy will be randomised and treated as one unit, that is, the patient will be randomised rather than the tendon. However an index tendon will be identified (this will be the one the patient perceives to be more severe at the point of randomisation).

## Randomisation

Study participants will be randomised strictly sequentially, as they become registered as eligible for randomisation on the telephone system. The treatment group will be allocated by computer using a minimisation algorithm and stratification by centre and laterality (one or both Achilles tendons affected) following a call to a secure, centralised, telephone-based randomisation service. Allocation concealment will be maintained by an independent randomisation team who will be responsible for generation of the sequence and will have no role in the allocation of participants. The randomisation system will allocate each participant a unique trial number.

Stratification by centre will help ensure that any clustering effect related to the centre itself will be equally distributed in the trial arms. Stratification on the basis of bilateral presentation will also be implemented to account for the poorer outcome associated with this sub-population.

## Post-randomisation withdrawals

Participants may withdraw from the trial treatment and/ or the trial at any time without prejudice. Unless a participant explicitly withdraws their consent, they will be followed-up wherever possible and data collected as per the protocol until the end of the trial. For participants explicitly withdrawing consent for follow-up procedures, trial data obtained up until the point of withdrawal will be included in the final analysis.

## Interventions

### Pre-injection

The PI at each site will identify relevant healthcare professionals to be trained and will record those who have completed training on the delegation log. Only those individuals listed on the delegation log will be allowed to prepare and administer trial interventions. At each centre the trial training programme will be delivered and documented by the chief investigator (Dr Kearney) and WCTU trial manager.

Following consent, completion of baseline questionnaires and randomisation, all participants will have approximately 10 mL of whole blood withdrawn from the antecubital fossa (vein at the elbow) and 5 mL of 2% lignocaine (local anaesthetic) will be injected into the skin overlying the painful tendon area for pain relief; this will be done with the participant in the prone (lying down and facing away) position on a treatment couch.

### PRP injection procedure

The whole blood will be centrifuged using the study specific Glo PRP system (Glofinn). Each centre will be supplied with the same centrifuge system to facilitate standardisation of the intervention.

Although treatment in the prone position means that the participant will be facing away from the healthcare professional, the syringe will be masked to make sure that the participant cannot see the contents of the syringe. Participants will then have one injection of the prepared platelet layer (approximately 3 mL) injected into the Achilles tendon using a standard 'peppering' technique at the site of the tendon pain. This technique involves a single skin portal and then five penetrations of the tendon.

### Placebo injection procedure

For the placebo injection, the masked needle will be inserted under the skin, but not into the tendon. The participant will feel the needle but nothing will be injected. There is an active debate pertaining to the treatment effect of needling trauma,[10] or the trauma of injecting fluid directly into the tendon. Therefore following discussion with the trial steering committee (TSC) and the data monitoring and safety committee (DSMC) a decision was made that a true placebo arm would need to avoid these possible treatment effects. The group consensus was therefore not to administer the placebo injection intratendinously and under the skin only. At the 6 months time point, following completion of all data collection, participants will be asked if they think they know their treatment allocation or not.

### Post-injection

After both the PRP and placebo treatments all participants will receive the same post-injection advice sheet. The post-injection advice sheet will inform participants that they may have increased pain for 24–48 hours, after which period they can resume their normal activities as pain allows. It will also include advice on potential adverse events (eg, infection and reddening of the skin) and what to do if they occur. Participants will be instructed not to undergo any further interventions for a period of 6 months, which will be monitored at each follow-up time point.

### Bilateral presentations

The index tendon will be randomised and managed accordingly. Regarding the non-index tendon the participant will have two options, to have no treatment or to receive a second injection into the non-index tendon.

### Technical failure

In the unlikely event that the project specific centrifuge system fails once blood has been drawn for PRP preparation/Placebo preparation the patient will be allocated to the placebo arm and analysed as a protocol violation.

### Quality assurance

Quality assurance checks will be carried out by a member of the trial team to assess compliance with the earlier intervention preparation and delivery. These checks will be done face to face and remotely using recording devices. Any deviations noted from the outlined trial interventions will be monitored by the oversight committees. If required, further training will be implemented to resolve any inconsistencies.

Additional quality assurance procedures to verify the quality of the PRP preparation will include research nurses from participating trial sites preparing PRP samples from healthy volunteers. Healthy volunteers will be recruited internally from these trial sites through advertisement in routinely distributed newsletters and posters. All volunteers will receive a screening phone call to be considered and only excluded if they have presence of systemic conditions (including diabetes, rheumatoid arthritis, peripheral vascular disease); they are pregnant and/or breastfeeding; use of anticoagulant therapy (warfarin, dabigatran, heparin) or are unable to adhere and consent to procedures.

Following consent procedures, healthy volunteers will provide two blood samples of up to 10 mL. Sample 1 will be kept as a whole blood control sample for analysis and sample 2 will be processed by the trial research nurses to produce a PRP sample for analysis. Samples will be anonymised and transported to an independent test lab which will conduct cell counts of whole blood and PRP preparations. Red cell, platelet and white cell counts (with full

differential count) using a blood counter will be undertaken. All samples will be destroyed after analysis.

## Blinding

It will not be possible to blind the research or clinical team involved in treatment preparation or delivery due to the nature of the intervention. All participants will be blinded and not know their treatment allocation through masking of the treatment syringe to prevent them from seeing the contents, as well as by being face-down during the injection. Participants in both groups will wait for approximately 30 min (PRP preparation time), and not be in the same room as the centrifuge. If this is not possible the participant will lie in the prone position and the centrifuge will spin without any PRP being prepared.

## Adverse event management

Adverse events will be listed on case report form for return to the 'ATM' office. Serious adverse events (SAEs) will be entered onto the SAE reporting form and reported to the central study team. All SAEs that occur between date of randomisation and the 2-week follow-up point will be followed-up as per protocol until the end of the trial. All SAEs will be reported to the sponsor (University of Warwick), ethics committee and oversight committees.

## End of trial

The trial will end when all participants have completed their 6-month follow-up. The trial will be stopped prematurely if mandated by the ethics committee, following recommendations from relevant oversight committees or funding for the trial ceases. The research ethics committee will be notified in writing within 90 days when the trial has been concluded or within 15 days if terminated early.

## Trial oversight

The trial management group (TMG) will meet monthly and consist of the project staff and co-investigators responsible for the day-to-day running of the trial. The TMG will report to an independent TSC who will hold meetings not less than once a year and take responsibility for approval of the protocol, monitoring and supervising the progress of the trial and considering recommendations from the DSMC. The TMG will also report to the independent DSMC, who will hold meetings not less than once a year. Confidential reports containing recruitment, protocol compliance, safety data and interim assessments of outcomes will be reviewed by the DSMC.

## Patient and public involvement (PPI)

PPI members have been active members on the TMG and TSC, and have reviewed all study materials. At the end of the study they will be contributing to dissemination plans distributed through the trial website and other relevant platforms.

## Statistical methods

The primary analysis will compare the VISA-A score at 6-month follow-up between the two treatment groups on an intention-to-treat (ITT) basis. Mixed-effects linear regression will be used with adjustment for design factors (centre and laterality), age, sex and baseline VISA-A score. Centre will be taken as a random effect to allow for possible heterogeneity in patient outcomes. Mean difference with 95% CI will be reported. Data transformation will be applied to the outcome before the modelling if VISA-A at 6 months is non-normally distributed. A p value<0.05 will be considered statistically significant.

The secondary analysis compares the outcomes including VISA-A at 3 months, EQ-5D-5L VAS and index scores at 3 and 6 months and pain VAS and complication rate at 2 weeks, 3 and 6 months. They will be analysed by ITT using linear regression with and without adjustment for design factors and baseline data. In addition, VISA-A at 6 months will be analysed in the following ways: (1) unadjusted linear regression by ITT and (2) adjusted mixed-effects regression by per-protocol to assess the impact of non-compliance. Kaplan Meier plot will be used to compare the time to complication by treatment if sufficient number is observed in each complication class. In case of bilateral tendinopathy (both sides), one randomly select side will be included in the analysis. Although missing data are not expected to be a problem for this study, the missing mechanism will be assessed and imputation may be used when deemed appropriate. The imputed data will be used for a sensitivity analysis. Prespecified subgroup analysis include laterality (single vs bilateral) and duration of symptom (≤median vs>median duration). The analysis will be adjusted in the same way as the primary analysis. A detailed analysis plan will be finalised prior to final data lock.

## Ethics and dissemination

PRP injections, among others, are currently used in clinical practice for mid-substance Achilles tendinopathy and do not expose trial participants to any substantial risks over and above standard care currently received.

The trial will be conducted in full conformance with the principles of the Declaration of Helsinki and to Good Clinical Practice guidelines and comply with all applicable UK legislation and Warwick University Standard Operating Procedures. All data will be stored securely and held in accordance with applicable data protection legislation. The trial will be reported in line with the Consolidated Standards of Reporting Trials (CONSORT) statement and minimum information for studies evaluating biologics in orthopaedics: PRP and mesenchymal stem cells.[20 21]

The results of this project will be disseminated through peer-reviewed journals, conference presentations among the orthopaedic and rehabilitation networks, policy makers such as NICE, patient-specific newsletters and through local mechanisms at all participating centres.

**Author affiliations**
¹Warwick Clinical Trials Unit, Warwick Medical School, University of Warwick, Coventry, UK
²Statistics and Epidemiology Unit, Warwick Medical School, University of Warwick, Coventry, UK
³Trauma and Orthopaedic Surgery, University Hospitals Coventry and Warwickshire NHS Trust, Coventry, UK
⁴Oxford Trauma, Nuffield Department of Orthopaedics, Rheumatology and Musculoskeletal Sciences, University of Oxford, Oxford, UK

**Acknowledgements** The research team would like to acknowledge Karen Keates for her contribution throughout the feasibility and main trial as a patient and public involvement representative.

**Contributors** RSK, MLC, NP and JY wrote the background section and developed the research question. RSK, MLC, NP and JB wrote the research methodology and management sections of the protocol. NP, CJ and JW wrote the sample size and statistical analysis sections of the protocol. All authors reviewed and approved the final manuscript.

**Funding** This trial was funded by VS Arthritis commencing 1 September 2015 (VS Arthritis 20831).

**Disclaimer** The funder has not been involved in the design of the study. The views expressed are those of the authors and not necessarily those of the National Health Service or the funder.

**Competing interests** None declared.

**Patient consent for publication** Not required.

**Provenance and peer review** Not commissioned; externally peer reviewed.

**ORCID iD**
Rebecca Samantha Kearney http://orcid.org/0000-0002-8010-164X

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
