## [Reviewer comments · BMJ Open]

ARTICLE DETAILS

TITLE (PROVISIONAL)	Platelet Rich Plasma versus Placebo for the management of Achilles Tendinopathy: Protocol for the UK study of Achilles Tendinopathy Management (ATM) multi-centre randomised trial
AUTHORS	Kearney, Rebecca; Parsons, Nicholas; Ji, Chen; Warwick, Jane; Brown, Jaclyn; Young, Jonathan; Costa, Matthew

VERSION 1 – REVIEW

REVIEWER	Shannon Munteanu La Trobe University, Australia
REVIEW RETURNED	23-Sep-2019

GENERAL COMMENTS	Thank you for allowing me to review this study protocol for a multi-site RCT investigating the efficacy of PRP injections for mid-portion Achilles tendinopathy. Overall the protocol is well written. I do have the following minor comments/questions: Introduction: P3L78: Please consider specifying the location of the pain is at the mid-portion of the Achilles (not just 'lower portion of the calf'). P3L90: A review of the previous trials in this area has been provided, and limitations provided. However, there still isn't a strong argument presented for the current trial (P4L105-6). What effect can the limitations from the previous trials have on the current evidence for effectiveness of PRP? How does the current trial improve on the previous trials and will this possibly lead to different findings? Methods and analysis: P6L143: To my knowledge, the MCID for the VISA-A for mid-portion AT isn't really known. Can the authors consider including the wording along the lines of '...estimated to lie between...' P5L159-68: How will the VISA-A be administered to non-athletic individuals (particularly question 8)? P5L166: please check 'compared'. General query regarding outcomes: What participant characteristics at baseline are being measured? Are you measuring use of co-interventions? P7L208: What diagnostic features will be used for ultrasound and MRI? Is the item 'if they have previously been randomised in the study' needed? Will you be including participants with MSK pathology of the lower limb (other than mid-portion AT)? If a participant has co-existing insertional AT, will they be included – please elaborate? P7L220: should 'were' be 'will'? P9L274-279: Will you be measuring if the blinding procedure has been successful? P9L277: This may be the first time that the abbreviations TSC and
---

	DMC have been used (ie they are spelt out on the subsequent page). Please can you check this. P11L368: The authors state that they will be analysing the ED5D VAS scores but previously (L173) it appears that the index value is the 'focus'. This creates some confusion – please can you be more clear. Can you please confirm if data substitution will be used for missing data (such as imputation) to be consistent with ITT (or will you be choosing to use it where 'appropriate'?) References: Please review the following: Reference 3 – spacing between word; Reference 4 – format of author section; Other comments The SPIRIT checklist with their submission. Can you please include a SPIRIT diagram (ie: see Figure 1: https://www.bmj.com/content/346/bmj.e7586).
--	--

REVIEWER	Rocco Papalia Campus Bio-Medico University of Rome - Italy
REVIEW RETURNED	12-Oct-2019

GENERAL COMMENTS	the manuscript represents a valuable contribution for research, although I think that the injection protocols should be revised, as one PRP injection may not sort an appropriate effect, as may be with a series of three injections.
--

REVIEWER	Paul Ackermann Karolinska Institutet, Dept of Molecular Medicine and Surgery, Stockholm, SWEDEN
REVIEW RETURNED	14-Oct-2019

GENERAL COMMENTS	Abstract Lines 57-58: “The first site opened to recruitment on 27th April 2016.”  - The reader wonders how many patients that has been included since April 2016? - Please include the number of patients included at the date of submission so that the readers can appreciate that the study has not ended? - Introduction Lines 98-100: “Of these four trials only one [12] has documented a significant treatment effect of PRP at 6 months (Mean VISA-A score 19.6 (SD = 4.5) in PRP group and 8.8 (SD =3.3) in placebo group).”  - Please elaborate more on why only one of four studies was successful in showing an effect of PRP? What was the methodological differences between the studies, which you will now use in the current study? Research Question Lines 116-120:
---

“In adults with painful mid-substance Achilles tendinopathy lasting longer than three months, does a single injection of PRP (intra-tendinous injection with a peppering technique) improve VISA-A (Victorian institute of Sport Assessment-Achilles) scores by a minimum of 12 points when compared to a placebo (subcutaneous injection of a dry needle and no peppering technique) injection at six months post injection?”

- Why is the peppering injection technique only used in the group receiving PRP and not in the dry needling group?
- If the hypothesis is that PRP improves healing, then all other interventions should be the same.

Materials and Methods:

Lines 152-156:

Outcome measures

“Outcome measures will be collected at baseline (pre randomisation) by a suitably qualified member of the research team, face to face at the recruiting site and then at two weeks by telephone direct from Warwick Clinical Trials Unit (WCTU) and three and six months post randomisation by postal questionnaire sent from WCTU or telephone if no postal response is received.”

- The outcome measures will be assessed in three different manners. At baseline the outcome measure will be assessed by face to face, then by telephone and then by postal response.
- Do you expect any problems with these three different assessments of the outcome?
- Will you present in each group (intervention and control) how many patients at each time-point that were assessed with each method?

Lines 273-279 Placebo Injection Procedure

For the placebo injection, the masked needle will be inserted under the skin, but not into the tendon. The participant will feel the needle but nothing will be injected. There is an active debate pertaining to the treatment effect of needling trauma [10], or the trauma of injecting fluid directly into the tendon. Therefore following discussion with the trial TSC and DMC a decision was made that a true placebo arm would need to avoid these possible treatment effects. The group consensus was therefore not to administer the placebo injection intratendinously and under the skin only.”

- This is a draw-back of the study that must be clearly presented in how the final results will be evaluated.
- In the final results you can not surely say that PRP is more effective than placebo.

Lines 381-385 Ethics and dissemination

“National Research Ethic Committee approved this study on 30th October 2015 (15/WM/0359). It was registered on the International Standard Randomised Controlled Trial Number registry with reference number ISRCTN 13254422 on 28th October 2015. The first site opened to recruitment 27th April 2016.”

	- Please include the number of patients included at the date of submission so that the readers can appreciate that the study has not ended?
--	---

VERSION 1 – AUTHOR RESPONSE

Reviewer: 1

1. Please consider specifying the location of the pain is at the mid-portion of the Achilles (not just 'lower portion of the calf').

This has been clarified to 'mid portion of the Achilles'

2. A review of the previous trials in this area has been provided, and limitations provided. However, there still isn't a strong argument presented for the current trial. What effect can the limitations from the previous trials have on the current evidence for effectiveness of PRP? How does the current trial improve on the previous trials and will this possibly lead to different findings?

The authors have clarified that none of the previous studies have evaluated the effectiveness of PRP across multiple sites, in an adequately powered trial, which this project will address.

3. To my knowledge, the MCID for the VISA-A for mid-portion AT isn't really known. Can the authors consider including the wording along the lines of '...estimated to lie between...'

The authors agree and have added this clarification

4. How will the VISA-A be administered to non-athletic individuals (particularly question 8)?

The VISA-A is being administered as per the developed and validated version, with no adaptations.

5. Please check 'compered'.

This has been amended to 'compared'.

6. General query regarding outcomes: What participant characteristics at baseline are being measured? Are you measuring use of co-interventions?

The following text has been added to clarify what patient characteristics are being collected at baseline and co-investigations:

'Baseline participant characteristics to be collected include duration of symptoms, side affected (left, right, bilateral), previous treatments for the Achilles tendinopathy, current medications, weight, height, age, gender, ethnicity, smoking and drinking status and employment status... Participants will be instructed not to undergo any further interventions for a period of six months, which will be monitored at each follow up time point.'

7. What diagnostic features will be used for ultrasound and MRI? Is the item 'if they have previously been randomised in the study' needed? Will you be including participants with MSK pathology of the lower limb (other than mid-portion AT)? If a participant has co-existing insertional AT, will they be included – please elaborate?

No diagnostic criteria for ultrasound/MRI diagnosis of tendinopathy were applied, to reflect routine clinical practice.

The item 'if they have previously been randomised in the study' is needed to ensure participants were only enrolled/randomised once into the study.

All the eligibility criteria used in the trial are outlined in the main text, under the heading 'screening and eligibility'. As outlined in this text participants with other MSK lower limb pathology were included unless: they have had prior Achilles tendon surgery or rupture on the index side, have had previous major tendon or ankle injury or deformity to either lower leg, if they have had a fracture of a long bone in either lower limb in the previous six months. Therefore participants with a coexisting insertional tendinopathy would be eligible for the study.

8. should 'were' be 'will'?

This has been changed.

9. Will be you be measuring if the blinding procedure has been successful?

We are collecting this data at the six month time point, and we have included a statement in the main text to clarify this point.

10. This may be the first time that the abbreviations TSC and DMC have been used (ie they are spelt out on the subsequent page). Please can you check this.

We have checked and amended accordingly.

11. The authors state that they will be analysing the ED5D VAS scores but previously (L173) it appears that the index value is the 'focus'. This creates some confusion – please can you be more clear. Can you please confirm if data substitution will be used for missing data (such as imputation) to be consistent with ITT (or will you be choosing to use it where 'appropriate'?)

Both the index values and VAS EQ 5D scores will be included as part of the secondary analysis, which is mentioned in both sections. As specified the focus is VISA-A, which is the primary.

12. Please review the following: Reference 3 – spacing between word; Reference 4 – format of author section.

These have been amended.

13. The SPIRIT checklist with their submission. Can you please include a SPIRIT diagram (ie: see Figure 1: <https://www.bmj.com/content/346/bmj.e7586>).

A table of the schedule of enrolment, interventions and assessments, in line with SPIRIT has been uploaded as a supplementary file (table two) and is referred to in the main text.

Reviewer: 2

1. The manuscript represents a valuable contribution for research, although I think that the injection protocols should be revised, as one PRP injection may not sort an appropriate effect, as may be with a series of three injections.

The authors are pleased that reviewer two feels this multi-centre RCT will be a valuable contribution for research. The team appreciate there are different PRP protocols and this will be a point of discussion in the main results paper.

Reviewer: 3

1. “The first site opened to recruitment on 27th April 2016.” The reader wonders how many patients that has been included since April 2016, please include the number of patients included at the date of submission so that the readers can appreciate that the study has not ended?

The authors have clarified that the trial is still ongoing with the following addition to the abstract text: ‘...the trial was in active recruitment at the point of submitting the protocol paper.’

2. “Of these four trials only one [12] has documented a significant treatment effect of PRP at 6 months (Mean VISA-A score 19.6 (SD = 4.5) in PRP group and 8.8 (SD =3.3) in placebo group).” Please elaborate more on why only one of four studies was successful in showing an effect of PRP? What was the methodological differences between the studies, which you will now use in the current study?

Previous RCTs in the area have been single site with small sample sizes, so are highly subject to type two error, this has been elaborated on in the main text.

3. “In adults with painful mid-substance Achilles tendinopathy lasting longer than three months, does a single injection of PRP (intra-tendinous injection with a peppering technique) improve VISA-A (Victorian institute of Sport Assessment-Achilles) scores by a minimum of 12 points when compared to a placebo (subcutaneous injection of a dry needle and no peppering technique) injection at six months post injection?” Why is the peppering injection technique only used in the group receiving PRP and not in the dry needling group? If the hypothesis is that PRP improves healing, then all other interventions should be the same.

The choice of comparator is justified in the text under the heading ‘Placebo Injection Procedure’.

4. “Outcome measures will be collected at baseline (pre randomisation) by a suitably qualified member of the research team, face to face at the recruiting site and then at two weeks by telephone direct from Warwick Clinical Trials Unit (WCTU) and three and six months post randomisation by postal questionnaire sent from WCTU or telephone if no postal response is received.” The outcome measures will be assessed in three different manners. At baseline the outcome measure will be assessed by face to face, then by telephone and then by postal response. Do you expect any problems with these three different assessments of the outcome? Will you present in each group (intervention and control) how many patients at each time-point that were assessed with each method?

The outcomes are patient completed at the baseline face to face and at the postal follow up time points of 3 and 6 months on paper CRFs, there are no differences in the collection of this data at these time points. The two week phone call does not collect any data beyond safety and pain, this will be reported in the final paper.

5. Procedure For the placebo injection, the masked needle will be inserted under the skin, but not into the tendon. The participant will feel the needle but nothing will be injected. There is an active debate pertaining to the treatment effect of needling trauma [10], or the trauma of injecting fluid directly into the tendon. Therefore following discussion with the trial TSC and DMC a decision was made that a true placebo arm would need to avoid these possible treatment effects. The group consensus was therefore not to administer the placebo injection intratendinously and under the skin only.” This is a draw-back of the study that must be clearly presented in how the final results will be evaluated. In the final results you can not surely say that PRP is more effective than placebo.

The choice of comparator and the interpretation of the results will be a discussion point in the final results paper.

6. “National Research Ethic Committee approved this study on 30th October 2015 (15/WM/0359). It was registered on the International Standard Randomised Controlled Trial Number registry with reference number ISRCTN 13254422 on 28th October 2015. The first site opened to recruitment 27th April 2016.” Please include the number of patients included at the date of submission so that the readers can appreciate that the study has not ended?

The authors have clarified that the trial is still ongoing with the following addition to the main text: ‘...the trial was in active recruitment at the point of submitting the protocol paper.’

VERSION 2 – REVIEW

REVIEWER	Shannon Munteanu La Trobe University, Melbourne, Australia
REVIEW RETURNED	11-Dec-2019
GENERAL COMMENTS	Thank you for replying to my queries.